# Fossil ribcages of *Homo sapiens* provide new insights into modern human evolution

José M. López-Rey [1,2] ✉, Isabelle Crevecoeur[3], Hila May [4,5], Dani Nadel [6], Carlos A. Palancar [1], Marta Gómez-Recio [1,2], Daniel García-Martínez[7,8,9] & Markus Bastir [1]

Recent research on the Nariokotome Boy's ribcage suggests the slender thorax of modern *H. sapiens* is a derived condition. However, since digital ribcage reconstructions of fossil *H. sapiens* are not available yet, it is unknown whether these individuals would have had a primitive or derived thorax. To address this issue, we first reconstructed the ribcages of Nazlet Khater 2, Ohalo II H2, Dolní Věstonice 13, and Ötzi. We used geometric morphometrics to compare them to 59 recent *H. sapiens* and three other *Homo* fossils (Nariokotome Boy, Kebara 2, Shanidar 3). Fossil *H. sapiens* ribcages exhibit the typical globular proportions of recent humans. Additionally, size and shape seem to be climate-dependent: smaller, cylindrical ribcages in warmer and more temperate climates (Nazlet Khater 2, Ohalo II H2) contrasted with larger, broader ribcages in colder climates (Dolní Věstonice 13). The ribcage of Ötzi presented mixed features, something that could have been beneficial for seasonal alpine transhumance. This suggests *H. sapiens* ribcage morphology encompasses both slender and stockier forms, highlighting that human anatomical variation might be more complex and context-dependent than previously thought.

Nowadays, the hominin fossil record has elucidated numerous inquiries about the postcranial evolution of the genus *Homo* up to *H. sapiens*[1]. There is, however, one major question concerning body proportions of modern humans, which is whether our slender and gracile Bauplan is a primitive or a derived condition in the *Homo* lineage. Initially, the first reconstructions of the most complete *H. erectus* skeleton discovered so far, KNM-WT 15000 or Nariokotome Boy, suggested a slender anatomy that would match ours[2]. The stocky and robust Bauplan associated with Neanderthals would be, therefore, apomorphic[3]. Nevertheless, recent research assessed a new ribcage reconstruction of the Nariokotome Boy and found it was shorter and stockier than previously proposed[4]. This finding contributed to others[5,6] and reversed the classical hypothesis suggesting that the slender *H. sapiens* Bauplan is a derived condition in the genus *Homo*, possibly in relation to climatic and locomotor adaptations[7].

The study of the hominin ribcage is, however, limited by the fact that ribs and vertebrae are very numerous and labile, so costovertebral series tend to appear incomplete, broken and/or commingled in osteo-archaeological collections[8–10]. Consequently, there have been only a few attempts of hominin ribcage reconstructions in the literature[3,4,11–14], none of them performed in fossil *H. sapiens*. This raises the question of whether the fossil *H. sapiens* ribcage resembled that of current humans or evolved within the species from a stockier to a slenderer – and more globular – configuration. In addition, the potential effects of climate on the Bauplan of fossil *H. sapiens* should be considered since recent research on current humans found latitudinal differences in ribcage size and shape[13].

The earliest *H. sapiens* specimens with costovertebral material to address this issue are Irhoud 14, Irhoud 15 (315 ± 34ky BP[15]), and Omo 1 (200ky BP[16]), all of them from the Middle Pleistocene. However, the first remains complete enough to provide an idea of the thoracic configuration of fossil *H. sapiens* are those from the Skhul and Qafzeh sites[17,18], dated to circa 100 ky BP. Unfortunately, their poor state of preservation —fragmentary, deformed ribs embedded in a block— and the ongoing debate regarding

[1]Paleoanthropology Group, Department of Paleobiology, Museo Nacional de Ciencias Naturales (MNCN-CSIC), Calle José Gutiérrez Abascal, Madrid, Spain. [2]Department of Biology, Faculty of Sciences, Universidad Autónoma de Madrid (UAM), Calle Darwin, 28049 Madrid, Spain. [3]UMR 5199 PACEA, CNRS, Université de Bordeaux, Pessac Cedex, France. [4]Department of Anatomy and Anthropology, Gray Faculty of Medical & Health Sciences, Tel Aviv University, Tel Aviv-Yafo, Israel. [5]Shmunis Family Anthropology Institute, the Dan David Center for Human Evolution and Biohistory Research, Gray Faculty of Medical & Health Sciences, Tel Aviv University, Tel Aviv-Yafo, Israel. [6]Zinman Institute of Archaeology, University of Haifa, Haifa, Israel. [7]Physical Anthropology Unit, Faculty of Biological Sciences, Universidad Complutense de Madrid (UCM), Calle José Antonio Novais, Madrid, Spain. [8]Center for Functional Ecology - Science for People and the Planet (CFE). Laboratory of Forensic Anthropology, Centre for Functional Ecology, Department of Life Sciences, University of Coimbra (UC), Calçada Martim de Freitas, Coimbra, Portugal. [9]Centro Nacional de Investigación sobre la Evolución Humana (CENIEH), Paseo de la Sierra de Atapuerca, Burgos, Spain. ✉e-mail: jolopezr@mncn.csic.es

their phylogenetic position[19,20] make it difficult to study them rigorously. From then on, there are other specimens dated during the Upper Pleistocene whose costovertebral skeleton could be of considerable interest. Nonetheless, ribcage reconstructions on these individuals are complicated due to their inaccessibility (e.g. Lake Mungo 3[21]) or because remains from different individuals are commingled (e.g., Cro-Magnon[22]). The aim of this research is, therefore, to gather a costovertebral sample of fossil *H. sapiens* from the Upper Pleistocene onwards to elucidate not only whether their reconstructed ribcages fall within the recent human range of variation, but also the potential relationship between their morphology and the location and ancestry of each specimen. For this purpose, we selected accessible skeletons from diverse locations, climates and ancestries whose costovertebral remains allow an accurate 3D ribcage reconstruction. These specimens are Nazlet Khater 2, Dolní Věstonice 13, Ohalo II H2, and Ötzi (Table 1).

## Nazlet Khater 2

Nazlet Khater 2 is the only complete adult skeleton from the MIS 3 found in North Africa, during a period when the climate was arid[23]. Dated on fragments of tooth enamel to ~38 ± 6 ky BP, this masculine individual was discovered in 1980 in an intentional burial associated with the Nazlet Khater 4 chert mining site near Tahta, Upper Egypt[24]. An anthropological study of this specimen was carried out by Crevecoeur[24,25], who described the presence of discrete characters shared with archaic *H. sapiens* and coupled with biomechanical adaptations to intense physical effort (e.g. high bone robusticity, pronounced muscular insertions), possibly due to mining work. This individual has been proposed to be part of a population of Sub-Saharan ancestry. In this scenario, Nazlet Khater 2 shows morphological features present in Middle/Late Pleistocene and Early Holocene Sub-Saharan specimens, which are absent in current African populations[26,27].

## Dolní Věstonice 13

The specimen Dolní Věstonice 13 is an exceptionally preserved masculine skeleton dated around 30 ± 1 ky BP and discovered in a triple burial during the excavations held in 1986 in Dolní Věstonice site (Moravia, Czech Republic)[28,29]. Beyond human remains, this site is also known for its rich archaeological findings, including art and tools that shed light on the lifestyle of modern humans in the harsh glacial environment of Central Europe during the Upper Paleolithic[30,31]. An anthropological study of Dolní Věstonice 13 revealed a combination of high bone robusticity and rare traits shared with the other two skeletons[32], whose kinship was confirmed after genetic analyses[28,33,34]. These publications also suggested a genetic admixture between Occidental and Oriental European human lineages in Dolní Věstonice individuals.

## Ohalo II H2

Discovered in 1989 near the Sea of Galilee (Israel) in a well-preserved context including artifacts and organic remains[35], Ohalo II H2 is the only complete adult masculine skeleton from the Upper Paleolithic found in this area. This specimen was dated to ~23 ± 0.5 ky BP[36,37], when climatic conditions in the Levant were cooler and more arid than they are today[38]. The importance Ohalo II H2 lies in the fact that it bridges the gap in thoracic fossils between early *H. sapiens* (Qafzeh and Skhul[17,18]) and Epipaleolithic semi-sedentary foragers (Natufians[39]) in the Levant. An anthropological description of this skeleton suggests a high bone robusticity and pronounced

muscular insertions in his upper limbs, likely resulting from foraging activities. However, similarities between these remains and those from the Natufian culture indicate trends of gracilization in these Levantine populations[36].

## Ötzi

Ötzi, 'the Iceman', is a Chalcolithic mummy discovered in 1991 in the Ötztal Alps, on the border between Austria and Italy[40,41]. Dated to ~5300 years ago, the study of Ötzi has revealed unique features such as tattoos[42], clothing and artifacts[43] that provide insights into his diet[44], lifestyle and health[45]. Furthermore, the skeleton of this mummy shows signs of high physical activity potentially caused by seasonal alpine transhumance[46]. A recent study of Ötzi's genome found a high proportion of Anatolian Neolithic farmer-related ancestry, distinguishing the Iceman from the lineage of Western European hunter-gatherers[47].

Knowing the background of these fossil *H. sapiens* specimens, our hypotheses are:

1. The slender *H. sapiens* Bauplan is a derived condition in the *Homo* lineage[4–6]. Hence, the four ribcages of fossil *H. sapiens* will be closer to recent *H. sapiens* than to other *Homo* fossils.
2. The relative size and shape of the *H. sapiens* ribcage is correlated to climate[13]. We, therefore, expect that Nazlet Khater 2 and Ohalo II H2 (warmer and more temperate climates) have smaller and slenderer ribcages compared to Dolní Věstonice 13 and Ötzi (colder climate).

## Results

Reconstructed ribcages of fossil *H. sapiens* exhibit the globular proportions typically described for recent humans (Fig. 1). Although there were no statistical size differences among any fossil *H. sapiens* and the comparative sample (Table 2), Nazlet Khater 2 and Ohalo II H2 were smaller and similar to temperate recent *H. sapiens*. Contrary, the ribcages of Ötzi and Dolní Věstonice 13 were larger and more comparable to cold-adapted *H. sapiens* and Neanderthals, respectively (Figs. 1 and 2). Considering mean values, the range of centroid sizes was bounded by non-*H. sapiens* fossils, with the Nariokotome Boy having the smallest value and Neanderthals the largest (Table 2).

In terms of shape, the permutations test did not find differences ($p > 0.05$) among any group at PC1 (23.3% of total variance), and PC2 (18.9% of total variance). Nonetheless, the distribution of the sample along these principal components is highly informative (Fig. 3). PC1 shows the relative relationship between ribcage width and depth (negative: width predominance; positive: depth predominance); and PC2 shows different degrees of rib declination (negative: higher declination; positive: lower declination). Interpreting the distribution of the PC scores reveals that the ribcages of Nazlet Khater 2 and Ohalo II H2 are not only similar in size but also in shape, as indicated by their PC1 and PC2 scores close to 0. This suggests a cylindrical ribcage shape, with proportional width and depth, together with medium rib declination in both specimens. Although Ötzi's PC1 and PC2 scores are similar to those of Nazlet Khater 2 and Ohalo II H2, his reconstructed ribcage is stockier with slightly higher rib declination, such as seen in Fig. 1. Among fossil *H. sapiens*, Dolní Věstonice 13 exhibits the widest ribcage with the lowest rib declination (PC1 scores < −0.05, PC2 scores > 0), being surprisingly similar to the Nariokotome Boy in PC1 and PC2.

**Table 1 | Comparative summary of the fossil *H. sapiens* chosen for this study**

| ID | Dating (ky BP) | Location | Climate | Ancestry | Specific physical activity |
|---|---|---|---|---|---|
| Nazlet Khater 2 | 38 ± 6 | North Africa | Warm | Sub-Saharan | Mining |
| Dolní Věstonice 13 | 30 ± 1 | Central Europe | Cold | Western and Eastern European | – |
| Ohalo II | 23 ± 0.5 | Levant | Temperate | Levantine | Foraging |
| Ötzi | 5.3 | Central Europe | Temperate - High mountain | Anatolian | Trashumance |

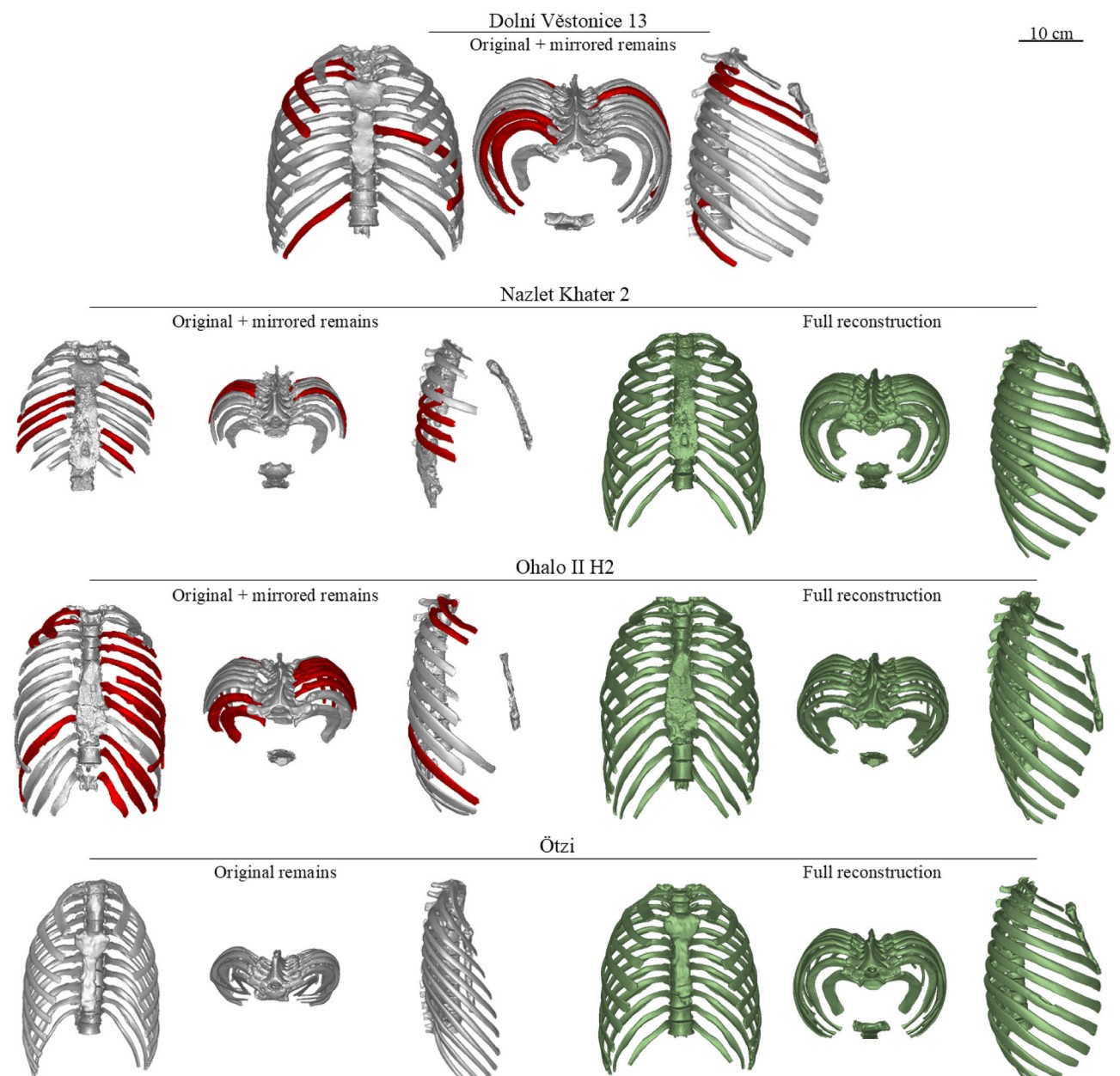

**Fig. 1 | Reconstructed ribcages belonging to fossil *H. sapiens*.** Color legend – Gray: original remains; Red: mirrored remains; Green: reconstructed ribcage.

**Table 2 | Statistical study of the centroid sizes per climatic group using permutations test (10,000 permutations)**

| Test | Statistics | Nazlet Khater 2 | Dolní Věstonice 13 | Ohalo II H2 | Ötzi | KNM-WT 15000 | Neanderthals | *H. sapiens* | | | |
|---|---|---|---|---|---|---|---|---|---|---|---|
| | | | | | | | | Cold | Temperate | Tropical | |
| Mean + Standard Deviation | Mean | 2580.6 | 2795.2 | 2603 | 2697.6 | 2466.1 | 2899.3 | 2642.2 | 2662.6 | 2470.8 | |
| | SD | – | – | – | – | – | 6.1 | 185.1 | 176.1 | 152.8 | |
| Permutations test | *p*-value | – | – | – | – | – | – | – | **0.04**\* | **0.003**\* | Neanderthals |
| | | – | – | – | – | – | – | **0.02**\* | **< 0.001**\* | - | *H. sapiens -* Tropical |

Only *p*-values <0.05 have been included. Significant *p*-values are marked with an asterisk (\*) and highlighted in bold.

## Discussion

This research describes the first reconstructed ribcages of fossil *H. sapiens* in order to elucidate whether they match those of recent humans and to explore the potential implications of climate and ancestry on their morphology. Figure 1 depicts the original and fully reconstructed thorax of Nazlet Khater 2, Dolní Věstonice 13, Ohalo II H2, and Ötzi, which exhibit the globular proportions typically described for *H. sapiens* in the literature[48–50]. Specifically, Nazlet Khater 2 and Ohalo II H2 share similar characteristics as both have small and cylindrical ribcages, resembling those recent *H. sapiens* living in temperate and warm areas. In contrast, thoracic

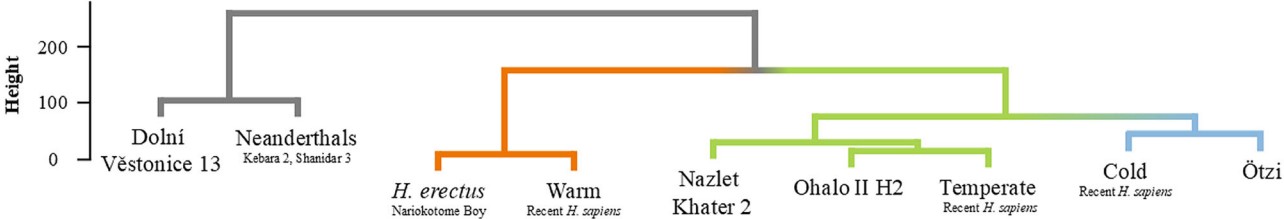

**Fig. 2 | Sample classification according to size.** Hierarchical cluster on the average (UPGMA) distances between the permuted centroid size of each group.

**Fig. 3 | Scatterplot showing the distribution of the principal components (PC)1, and PC2 scores extracted from the PCA in shape space.** The accompanying visualizations show the shape of the ribcage at the extremes of the axes.

proportions are larger and wider in Ötzi, and especially in Dolní Věstonice 13. This latter specimen is particularly striking because, while the size of its ribcage is comparable to that of Neanderthals, its shape is similar to KNM-WT 15000 (Table 2, Figs. 2 and 3). It is important to note that the Nariokotome Boy is a juvenile individual, and its ribcage morphology may not fully represent adult anatomy[4]. This could influence interpretations of morphological similarity, particularly when comparing it to adult specimens like Dolní Věstonice 13. Additionally, the small size of the KNM-WT 15000 ribcage is consistent with its immature age. Considering this, our findings on fossil *H. sapiens* can be interpreted in terms of paleoecology and ancestry of the chosen specimens.

To begin with, López-Rey et al.[13] found that Allen and Bergmann's rules apply to the recent *H. sapiens* ribcage. Thus, individuals from warmer and more temperate climates might have smaller and slenderer ribcages compared to others from colder areas, such as found for Nazlet Khater 2 and

Ohalo II H2 compared to Dolní Věstonice 13. The situation of Ötzi is relatively ambiguous since his ribcage is larger than that of Nazlet Khater 2 and Ohalo II H2 but shares with them a similar shape. These mixed thoracic features could have been beneficial for someone like Ötzi, who lived during periods of alternating residence between the southern, temperate European lowlands, and the Alpine region[46]. In addition, implications of paleoenvironment in the anatomy of these individuals might be confirmed by their ancestry (Table 1). None of the four studied fossils share the same genetic stock since 1) Nazlet Khater 2 has been proposed to belong to a population of Sub-Saharan ancestry[26,27]; 2) Dolní Věstonice 13 shows an admixture of Western and Eastern European genes[33,34]; 3) Ohalo II H2's population was presumably native from the Levant[36]; and 4) the genome of Ötzi presents a high proportion of genes from Anatolian populations that migrated to Europe spreading farming[47]. Knowing this, anatomical resemblance —or differences— found among individuals might be supported by climatic

https://doi.org/10.1038/s42003-025-08472-3                                                                                          **Article**

adaptations rather than any genetic proximity between them. If happening according to this premise, these climatic adaptations might be especially strong in specimen Dolní Věstonice 13.

As commented above, the thoracic proportions of Dolní Věstonice 13 are remarkable because his ribcage is almost as large as that of Neanderthals, but relatively wider, more similar to the Nariokotome Boy. Previous research on this topic proposed stockier and bigger ribcages for those *H. sapiens* adapted to colder climates in order to retain heat[13]. According to the literature, Dolní Věstonice 13 inhabited Central Europe during the Last Glacial Maximum, when climatic conditions were extreme[31]. Thus, such thoracic proportions could have been beneficial for surviving that hostile climate. Geometrical affinities found between Dolní Věstonice 13 and other fossil *Homo* from warmer climates (Kebara 2, Shanidar 3, and KNM-WT 15000) might be a matter of equifinality —an anatomical convergence driven by different evolutionary paths[51]. This assumption could be supported by evidence from other specimens such as those found in Sima de los Huesos, Spain (*H. heidelbergensis*/early Neanderthals, 430 ky BP[52]), or Gona, Ethiopia (*H. erectus* s. l., BSN49/P27, 0.9–1.4 My BP[53]). While no ribcage reconstructions are available for these individuals, the latest descriptions of their remains suggest they likely had wide and deep ribcages, even though they did not inhabit extremely cold ecosystems. These observations open the door to new research on the role of climate in the morphological evolution of hominins, especially regarding what other factors could have driven the development of a stocky Bauplan in temperate and warm environments.

Besides, there are interesting points to discuss on the statistical interaction between the four *H. sapiens* fossil ribcages and the comparative sample. First, size and shape variability in recent *H. sapiens* comprises almost that of fossil individuals (Fig. 3, Table 2). Consequently, statistical analyses were not significant. This might be also subjected to the methodological limitations of the reconstruction protocol and geometric morphometrics. Primarily, methods of estimation such as vectors of change and partial least squares (PLS) analysis embedded structural assumptions derived from the modern human reference sample. These assumptions could have subtly influenced aspects of the reconstructed morphology, such as rib orientation, curvature, or thoracic volume, especially in cases where preservation is very limited. Although estimations were conducted using as much of the original fossil material as possible, differences between the control sample and the fossil reconstructions may have become attenuated, and some degree of uncertainty remains when interpreting the results. Furthermore, by reducing the full anatomy of the ribcage to the geometry of specific (semi)landmarks, certain distinctive features—such as robustness or muscle attachment areas—that are evident during visual inspection of the fossils are not captured in the statistical analyses[54,55]. As a result, the Neanderthal ribcage shape, as represented by the analyzed PC scores, may appear more similar to that of *H. sapiens* than it actually is when considering discrete anatomical traits[3,11,12,14,55,56]. In fact, a mere observation to Fig. 1 confirms that the four reconstructed ribcages are unequivocally from *H. sapiens* given their globular shape[4,11,13,48,49].

Nevertheless, if we strictly interpret our statistical results, Hypothesis 1 has to be rejected. This Hypothesis stated that the ribcage of fossil *H. sapiens* would be different from *H. erectus* and Neanderthals given the (supposedly) derived slender Bauplan of *H. sapiens*. In our results, the ribcages of fossil *H. sapiens* are not necessarily similar to recent modern humans in terms of size and shape (see the case of Dolní Věstonice 13). In contrast, our results support Hypothesis 2 as they may indicate that ribcage morphology in fossil *H. sapiens* would be influenced by climate. This research joins others[1,13,57] and suggests a great morphological variability in *H. sapiens* sensu lato (both fossil and recent) that cannot be reduced to a generalized slender Bauplan, as it depends not only on genetics but also on the climatic plasticity of our species. Apparently, slender ribcages are exclusive from *H. sapiens*, but stockier ribcages such as those found in other *Homo* species are also encompassed in *H. sapiens* morphological variability. Consequently, (geo)metric measurements are informative but not definitive, and discrete features[55] proven to be the key in distinguishing hominins in ribcage studies.

## Conclusions

Reconstructed ribcages of fossil *H. sapiens* have the globular proportions typically described for modern humans. Besides, those fossil *H. sapiens* from warmer and more temperate areas have smaller and slenderer ribcages compared to others inhabiting colder regions such as Dolní Věstonice 13, whose thorax is particularly large and stocky, similar to Neanderthals in size and KNM-WT 15000 in shape. This suggests a great morphological variability in *H. sapiens* that cannot be reduced to a general Bauplan as body proportions might depend not only on genetics but also on the climatic plasticity of our species. Slender ribcages seem to be unique to *H. sapiens*, but stockier ribcages, like those found in other *Homo* species, are also part of the variation seen in both fossil and recent *H. sapiens*.

## Materials and methods
### Sample description
Individuals studied in this article can be divided into three categories: fossil *H. sapiens*, recent *H. sapiens* and other fossil *Homo*. The first group includes Nazlet Khater 2, Dolní Věstonice 13, Ohalo II H2, and Ötzi, adult masculine fossils whose general information has been detailed above and is summarized in Table 1. In order to build a robust comparative sample (Table 3), we gathered the previously reconstructed ribcages of 59 recent adult *H. sapiens* males[13], all of them deposited in recognized museums except those individuals requested to the New Mexico Decedent Image Database (NMDID), who are recently deceased. The ribs and vertebrae of all these individuals had no pathological or taphonomical alterations that could affect the overall ribcage shape. These skeletons were chosen from 19 populations distributed along the five inhabited continents and grouped according to the average annual temperature of their location in cold (≤7 °C), temperate (≤23 °C) and warm (>23 °C) populations. In addition, we studied the ribcages of three fossil *Homo*: KNM-WT 15000 (Nariokotome Boy, juvenile *H. erectus*, 1.5 My BP), Kebara 2 (adult *H. neanderthalensis*, 60 ky BP), and Shanidar 3 (adult *H. neanderthalensis*, 50 ky BP), digitally reconstructed by Bastir et al.[4], Gómez-Olivencia et al.[11] and López-Rey et al.[14], respectively. Before proceeding with the fossil *H. sapiens* reconstructions, we measured the already reconstructed ribcages of the comparative sample in Viewbox v. 4.1 (https://www.dhal.com/viewbox.htm) using a template of 526 (semi)landmarks (Supplementary Fig. 1, Supplementary Table 1), which was previously described by Bastir et al.[50]. The obtained 3D coordinates will be required afterwards.

### Costovertebral preservation in fossil H. sapiens
We evaluated the state of preservation of the ribs and vertebrae belonging to each fossil *H. sapiens* right before starting with each reconstruction protocol. On the one hand, all costovertebral levels are present and complete in both Dolní Věstonice 13 and Ötzi except rib 12, which is fragmented in Dolní Věstonice 13[32] and vestigial in Ötzi[58]. Nevertheless, contrasted to the unaltered ribs of Dolní Věstonice 13, the taphonomic effects of Ötzi's prone position bent his ribs dorso-ventrally[41,58]. On the other hand, although mostly all costovertebral levels are present in Nazlet Khater 2 and Ohalo II H2, only several are fully preserved. Specifically, vertebrae T1–T5 in Nazlet Khater 2[24]; and ribs 1, 7, 9-12 and vertebrae T1, T2, T3, T8 in Ohalo II H2[36].

### Ribcage reconstructions in fossil H. sapiens
Among all fossil *H. sapiens* individuals, Dolní Věstonice 13 was the only one whose costovertebral material was complete in both number and conservation of elements. Hence, we digitally reconstructed the ribcage of this specimen following the method standardized by López-Rey et al.[59] using the software LhpFusionBox (Université Libre de Bruxelles). To summarize, we first calculated the theoretical thoracic kyphosis of this specimen using the 'thoracic vertebral body height difference' (TVBHD[60]) method. This method consists in measuring the anterior (A) and posterior (P) vertebral body heights from T1 to T12 and dividing their sum to obtain the A–P ratio. Then the A–P ratio is included in a regression formula whose result is the approximate angle of thoracic kyphosis. Following, we reconstructed the thoracic spine by the corrected 'zygapophyseal facet method' (ZAM[61]),

**Table 3 | General information about the comparative sample**

| Category | ID | Population | Site | n | Latitude (approx.) | Host institution |
|---|---|---|---|---|---|---|
| Recent *H. sapiens* (Cold) | 99.1 400/481/542 | Inuit | Point Hope, Alaska (U.S.A.) | 3 | 70° | American Museum of Natural History (N.Y.C., U.S.A.) |
| | KAL 0041/0151/0152 | Inuit | Greenland (Denmark) | 3 | 68° | University of Copenhagen (Denmark) |
| | KAL 0924/0926 | Norsemen | Nuuk, Greenland (Denmark) - Scandinavian origin | 2 | 64° | |
| | 417/418/ED 10/Las Mandibulas | South Patagonian - Fuegian | Isla Grande de Tierra del Fuego (Argentina/Chile) | 4 | −53° | Universidad del Centro de la Provincia de Buenos Aires (Argentina)/ Museo Nacional de Historia Natural de Chile (Santiago, Chile) |
| Recent *H. sapiens* (Temperate) | 34982/34983/35014 | French | Paris (France) | 3 | 48° | Musée de l'Homme (Paris, France) |
| | ACC 27/58/77 | Spanish | Madrid (Spain) | 3 | 40° | Universidad Complutense de Madrid (Spain) |
| | 119259/170124/184880 | Native North American | New Mexico (U.S.A.) | 3 | 35° | New Mexico Decedent Image Database (NMDID) |
| | 23873/23867 | Native North American | California (U.S.A.) | 2 | 34° | Musée de l'Homme (Paris, France) |
| | 20820/24990/24985 | Japanese | Hyōgo Prefecture (Japan) | 3 | 34° | |
| | 11247/12245/12246/11248/11249 | Native Canarian | Gran Canaria, Canary Islands (Spain) | 5 | 28° | |
| | 106127/124031/161103 | Mexican | Mexico | 3 | 25° | New Mexico Decedent Image Database (NMDID) |
| | 25472/25478/25463 | Native South American | Santiago del Estero (Argentina) | 3 | −27° | Musée de l'Homme (Paris, France) |
| | 12262/25503/12260/12267 | North Patagonian | Rio Negro and Chubut (Argentina) | 4 | −41° | |
| Recent *H. sapiens* (Warm) | 112/166/192 | Nubian | Abri, Northern State (Sudan) | 3 | 21° | Universidad Complutense de Madrid (Spain) |
| | 8274/8278/8277 | Negrito | Luzon (Philippines) | 3 | 15° | Musée de l'Homme (Paris, France) |
| | 9778/9779/9776 | Papuan | New Britain (Papua New Guinea) | 3 | −6° | |
| | 23943/19855/19852 | Native South American | Ancón (Peru) | 3 | −9° | |
| | 25/77/107 | Sub-Saharan | Lagos (Portugal) - Subsaharan origin | 3 | 15° to −20° | DRYAS Octopetala (Coimbra, Portugal) |
| | 100139/103805/111819 | Afro-American | New Mexico (U. S. A.) - Subsaharan origin | 3 | 35° | New Mexico Decedent Image Database (NMDID) |
| Other fossil *Homo* | KNM-WT 15000 | African *H. erectus* | West Turkana (Kenya) | 1 | 4° | Kenya National Museum (Nairobi, Kenya) |
| | Kebara 2 | Levantine Neanderthal | Kebara Cave (Israel) | 1 | 32° | Tel Aviv University (Israel) |
| | Shanidar 3 | Levantine Neanderthal | Shanidar Cave (Iraq) | 1 | 36° | Smithsonian Institution (Washington D.C., U.S.A.) |

which consists in respecting contiguity and maximal overlap of the zyga-pophyseal facets taking into consideration the wedging of the vertebral body. Once the spine was reconstructed, we located ribs in their corresponding place, respecting their expected position in functional residual capacity (FRC[62]), and then measured the 3D form of the obtained ribcage in Viewbox v. 4.1. using the abovementioned template.

Remains belonging to Nazlet Khater 2, Ohalo II H2 and Ötzi were reconstructed following a more complicated methodology since their costovertebral levels are mostly present, but almost all of them are damaged or altered. To begin with, when necessary, we restored the thoracic vertebrae of these individuals following protocol developed by García-Martínez et al.[63]. To synthetize, we first scanned by Artec Spider (www.artec3d.com) the vertebral material belonging to a control sample of 10 adult *H. sapiens* males housed at the Faculty of Medicine of the Universidad Complutense de Madrid. We then quantified the size and shape of these scans using a specific template. Next, we calculated the vector of change between each mean metamer as the increment of 3D coordinates between each vertebral level and the subsequent one. This vector was added or subtracted to the best-preserved vertebral levels of fossil *H. sapiens* depending on whether we wanted to estimate missing elements below or above them, respectively. Eventually, the obtained coordinates were warped, and the final meshes were obtained adding the preserved fossil material to the estimated volumes. The thoracic spine of Nazlet Khater 2, Ohalo II H2 and Ötzi was subsequently reconstructed and their costal remains placed where it corresponds according to López-Rey et al.[59].

To estimate the full length of each rib, we developed a new method based on a partial least squares (PLS) analysis. Firstly, we quantified the size and shape of the incomplete ribcages of each fossil *H. sapiens* using the template described by Bastir et al.[50], but only taking into consideration those (semi)landmarks present in the three original skeletons. Then we used the 59 recent *H. sapiens* individuals to perform the PLS analysis between two blocks: full ribcage vs partial ribcage – containing those (semi)landmarks present in each partial fossil. Next, we performed a GPA in form space per block and aligned the case —that is, the (semi)landmarks of each partial fossil ribcage— to the block where we selected those preserved landmarks in the control sample. Then we performed the PLS analysis between the two blocks and predicted on one block (full ribcage) the case aligned to the other (partial ribcage) using the first latent variable of the analysis. Consequently, the full ribcage of Nazlet Khater 2, Ohalo II H2 and Ötzi was estimated. Ribcage reconstructions of fossil *H. sapiens* using original and restored material can be visualized in Fig. 1.

### Statistics and reproducibility

Before proceeding with the statistical analyses, we symmetrized each ribcage and removed the coordinates from ribs 11 and 12 since these costal levels are very variable in form[64]. This procedure avoids the effect of morphological alterations and reduces noise in the analysis and interpretation of the results. The effects of scale, position and rotation among the final 3D coordinates of the full sample were removed by performing a generalized Procrustes analysis (GPA[65]). General size differences among the sample were explored using a permutations test (10,000) on the centroid size (Table 2), which is a measure of scale for (semi)landmark configurations, calculated as the square root of the summed squared deviations of the coordinates from their centroid—the coordinate-wise average of the (semi)landmarks of one form[65]. Results were depicted in Fig. 2 by a hierarchical UPGMA cluster analysis[66], a method that groups samples based on the average distances between the permuted centroid sizes of each group. General shape differences were tested by a principal component analysis (PCA) in shape space. Differences among groups were statistically checked using permutations tests (10,000 permutations) and visualized by a scatterplot on PC1-PC2 (Fig. 3). Supplementary Data 1 and Supplementary Data 2 provide the information for plotting Figs. 2 and 3, respectively. All the statistical analyses were performed in RStudio v. 2023.12.1-402[67] using the packages 'FSA' v. 0.9.5[68], 'geomorph' v. 4.04[69], 'ggplot2' v. 3.3.3[70], 'Morpho' v. 2.10, 'Rvcg' v. 0.22.1[71], and 'stats' v. 4.2.3[67]. The full R script is openly available[72].

### Ethical statement

This study used human osteoarchaeological remains from documented collections housed in accredited institutions, with appropriate permissions from curators. All analyses were non-invasive and conducted in accordance with ethical guidelines and institutional standards. We recognize the cultural and historical significance of these remains and we are committed to treating them with respect and scientific integrity.

### Reporting summary

Further information on research design is available in the Nature Portfolio Reporting Summary linked to this article.

### Data availability

Costovertebral image data must be requested from the corresponding housing institutions and curators. Other kinds of data that support the findings of this study are available from the corresponding author to any researcher for purposes of reproducing the analyses. Numerical source data for plotting the UPGMA cluster and the PCA is provided in Supplementary Data 1 and Supplementary Data 2, respectively.

### Code availability

The R script, including all analyses and figure preparations, is deposited in Zenodo and is freely available (https://doi.org/10.5281/zenodo.15753026).

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

## Acknowledgements

We would like to thank the following researchers and technicians for their help in accessing and scanning the recent *H. sapiens* sample: Prof. Ashley Hammond, Dr. Niels Lynnerup, Dr. Chiara Villa, Dr. Martin Friess, Liliana Huet, Véronique Laborde, Prof. Bernardo Perea, Dr. Maria Teresa Ferreira, Dr. Miguel Almeida, Dr. Luis Ríos, Jorge Sanz and Mar Casquero. In addition, we acknowledge the following researchers for allowing us the access and study of the fossil material: Dr. Matthew Tocheri for Shanidar 3; Prof. François Bon, Dr. David Pleurdeau, Dr. Joséphine Lesur and Dr. Chantal Tribolo for Nazlet Khater 2 (micro-CT-scanned thanks to the ANR project "Big Dry", reference: ANR-14-CE31); Dr. Wolfgang Recheis for Ötzi; and Dr. Sandra Sazelova for granting access and hosting during the scanning of Dolní Věstonice 13. Regarding funding information, grant PRE2021-097584 to JMLR and grant PID2020-115854GB-I00 to MB are funded by MCIN/AEI/10.13039/501100011033 of the Spanish Ministry of Science and Innovation and the European Union. The work of DGM is carried out at the R&D Unit Center for Functional Ecology - Science for People and the Planet (CFE), with reference UIDB/04004/2020, financed by FCT/MCTES through national funds (PIDDAC) (DOI 10.54499/UIDB/04004/2020).

## Author contributions

Conceptualization: J.M.L.R., M.B. Data curation: I.C., H.M., D.N. Investigation: J.M.L.R., D.G.M., M.B. Methodology: J.M.L.R., C.A.P., M.G.R., D.G.M., M.B. Formal analysis: J.M.L.R., C.A.P., M.G.R. Funding acquisition: D.G.M., M.B. Project administration: D.G.M., M.B. Software: J.M.L.R., C.A.P., M.G.R. Resources: J.M.L.R., C.A.P., M.G.R., D.G.M., M.B. Visualization: J.M.L.R., M.G.R. Validation: J.M.L.R., D.G.M., M.B. Supervision: I.C., H.M., D.N., C.A.P., M.G.R., D.G.M., M.B. Writing—original draft: J.M.L.R. Writing—review & editing: J.M.L.R., I.C., H.M., D.N., C.A.P., M.G.R., D.G.M., M.B.

## Competing interests

The authors declare no competing interests.
