## [Transparent Peer Review file · Communications Biology]

Fossil ribcages of Homo sapiens provide new insights into modern human evolution

Corresponding Author: Mr José López-Rey

Version 0:

Reviewer comments:

Reviewer #1

(Remarks to the Author)

1. General summary. This manuscript presents a concise analysis of four newly articulated and digitized hominin rib cages and thoracic vertebrae in comparison to a world-wide set of recent humans and three previously analyzed hominids (KNM WT-15000, attributed to Homo erectus plus Shanidar 3 and Kebab 2, which are both attributed to Homo neanderthalensis. The authors provide a novel yet convincing methodology for imputing missing skeletal elements in some of the new fossils. Their analysis shows that many modern humans and Homo erectus have medio-laterally narrow and "globular" rib cages while Neanderthals have a transversely wider thorax. A novel finding that the authors report is that Dolni Vestonice 13, a Gravettian early modern human from Czechia, also has a transversely broad rib cage as do some cold-adapted recent humans. They conclude that this variation likely represents an example of climatic (ecogeographic) variation rather than a phylogenetic character that distinguishes Neanderthals from H. erectus and H. sapiens.

I found the article well written, clear, and think it will be of interest to the scientific community. I recommend the manuscript be accepted, but have a few points I think the authors should consider that might improve the paper. I do not think that any of my criticisms should be viewed by the Editor or authors as serious flaws.

2. Comments to the authors.

The paper might be improved by a brief discussion of the implications of the Sima de los Huesos pelvis, which suggest a wide lower thorax (and possibly all of the thorax at ~400 ka. It might also be worthwhile to mention the Gona pelvis, which if it is Homo erectus, would suggest relatively wider hips than proposed for KNM-WT 15000 by Ruff and Walker. In particular, the Gona pelvis might further complicate the ecogeographic story for thorax morphology.

The method for estimating the position, size, and orientation of missing elements described on lines 285-297 raised some worries in my mind about how much error the procedure would produce in the resulting estimates. Can the authors provide a few more statistical details to clarify the magnitude of the uncertainty? I worry that the (very lovely) graphics in Figure 1 look authoritative but may conceal more uncertainty than they imply.

I have some more minor comments that are described below by line number of the review copy of the manuscript.

Line 82: Please note that the new age for Jebel Irhoud reported by Dublin et al. is 315 ± 34 ka.

Lines 123-134: Perhaps specify that Ohalo 2 bridges the gap in thoracic fossils between Skull/Qazeh and later modern humans in the Levant; there are additional, earlier cranial fragments of modern humans (e.g., the calvaria from Manot Cave).

Line 639: Please change (Irak) to (Iraq).

Reviewer #2

(Remarks to the Author)

Thank you for the opportunity to review the manuscript entitled "Ancient ribcages, modern form? New insights into Homo sapiens evolution".

This study examines the ribcage morphology of fossil H. sapiens to assess the primitive or derived status of the modern H. sapiens thorax. The authors find that ribcage morphology in fossil H. sapiens is more variable than previously suggested and that size and shape are climate dependent.

Overall, the manuscript is well-written and engaging. The authors provide solid background information, particularly regarding the specimens used in their research, which effectively sets the stage for their analyses. While most of my comments are minor clarifications, concerns regarding the reconstruction methodology and interpretation of the shape plot warrant more significant consideration and could impact the results and main discussion points:

1. Ohalo II H2 is categorised within the 'warmer climate' sample in line 142, yet the background information on lines 122-123 states that "climatic conditions in the Levant were cooler and more humid than they are today". Additionally, Table 1 lists the climate for this specimen as temperate. Please clarify.

2. The term 'extant' is used in Figure 1, line 302, and line 307 in reference to fossil H. sapiens specimens, which creates confusion since 'extant' typically refers to living species. Using 'original' vs 'reconstructed' material may perhaps provide greater clarity.

3. The introduction states that the Skhul and Qafzeh ribcages are sufficiently preserved, yet they are not included in this study. What is the reason for their exclusion?

4. Regarding the Nariokotome Boy, was the juvenile reconstruction or the estimated adult ribcage from Bastir et al. (2020) used in the analyses?

5. An image of the (semi)landmarks used in the analyses would be beneficial, along with a brief description of the (semi)landmark placement. This is particularly relevant given the discussion on reducing full ribcage anatomy to landmark data.

6. The fossil reconstructions of Nazlet Khater 2, Ohalo II H2, and Ötzi are based on 10 recent male H. sapiens specimens. By using the spatial arrangement of recent H. sapiens ribcages for the reconstructions, these reconstructions will inevitably inherit some of the characteristics of the recent H. sapiens ribcages, such as its globular proportions. The shape plot underscores this issue, as the only ribcage that stands out from the modern H. sapiens means is Dolní Věstonice 13. Would using complete fossil ribcages as templates for reconstructing the three distorted fossil H. sapiens specimens result in different reconstructions?

7. The centroid size results are well-written and clear. However, the analysis requires some more explanation in the methods section.

8. Dolní Věstonice 13 clearly stands out from the other three fossil H. sapiens in shape space, while Nazlet Khater 2, Ohalo II H2, and Ötzi cluster more centrally on the shape plot (perhaps due to the reconstruction method as noted above). Hence, grouping Dolní Věstonice 13 and Ötzi as exhibiting 'colder' morphologies while classifying Nazlet Khater 2 and Ohalo II H2 as 'warmer' morphologies is not supported by the shape analysis. More justification for this grouping is needed beyond the brief mention in the results section that the Ötzi reconstruction displays a stockier and slightly higher rib declination.

9. Lines 213-215: The wording here is awkward and unclear. Please revise.

With these considerations in mind, I believe the manuscript has the potential to make a meaningful contribution to the field. Addressing these concerns will enhance the clarity and robustness of the findings.

Version 1:

Reviewer comments:

Reviewer #1

(Remarks to the Author)

Review of revised manuscript "Ancient ribcages, modern form? New insights into Homo sapiens evolution," by José López-Rey et al. (COMMSBIO-25-1445A).

This revised manuscript has successfully addressed all of my criticisms of the first version. I found no additional problems.

I advise the Editor to ACCEPT the manuscript.

Reviewer #2

(Remarks to the Author)

Thank you for sending the revised manuscript "Ancient ribcages, modern form? New insights into Homo sapiens evolution". Overall, the authors did a great job of revising this manuscript and I commend the clarifications provided. However, a few outstanding issues remain that should be addressed before publication.

Firstly, I appreciate the authors' response regarding the use of modern human samples in fossil ribcage reconstructions. The clarification that recent *H. sapiens* data were used specifically for estimating missing vertebral elements and rib lengths, rather than for global ribcage morphology, is helpful. However, I believe the inherent limitations of the approach should be addressed somewhere in the manuscript. Even when used in a localised way, methods such as vectors of change and PLS embed structural assumptions derived from the modern human sample. These assumptions could subtly influence aspects of the reconstructed morphology, such as rib orientation, curvature, or thoracic volume, especially in cases where preservation is very limited. It would be helpful to explicitly acknowledge this in the discussion or methods section, along with the authors' efforts to preserve as much of the original fossil morphology as possible. Additionally, a brief discussion on whether the use of complete fossil ribcages as reference templates (where available) might yield meaningfully different reconstructions would be appropriate.

Secondly, the authors' clarification concerning the inclusion of Ohalo II H2 in the "warmer climate" group provides helpful context. However, I still find this classification potentially confusing, especially given that the manuscript refers to the climate of Ohalo II H2 as "cooler and more humid than today" and labels it as "temperate" in Table 1. I suggest that the manuscript be revised to either (1) clearly explain the rationale for grouping Ohalo II H2 with the "warmer climate" specimens, despite its temperate classification, or (2) adjust the grouping terminology to avoid potential misinterpretation.

Thirdly, since the reconstruction of KNM-WT 15000 used in the analyses represents a juvenile individual, this should be acknowledged in the discussion, especially when highlighting the proximity of Dolní Věstonice 13 to this specimen. The ribcage morphology of a juvenile may not reflect that of an adult, which could influence interpretations of morphological similarity. Given its immature age, it is also not surprising that KNM-WT 15000 has the smallest ribcage in the sample, and this should be acknowledged in the manuscript.

Lastly, the phrase on line 175-176: "Fig. 1 depicts the extant and fully reconstructed thorax of Nazlet Khater 2, Dolní Věstonice 13, Ohalo II H2, and Ötzi" should be revised to replace "extant" with "original" in line with the rest of the revised manuscript terminology.

RESPONSE LETTER

Before starting to respond to your comments, the coauthors and I would like to thank the reviewers for investing their time in reading the manuscript and sending us their opinion. We hope that our responses and modifications would satisfy their inquiries.

REVIEWER 1

Major comments

COMMENT 1: The paper might be improved by a brief discussion of the implications of the Sima de los Huesos pelvis, which suggest a wide lower thorax (and possibly all of the thorax at ~400 ka). It might also be worthwhile to mention the Gona pelvis, which if it is *Homo erectus*, would suggest relatively wider hips than proposed for KNM-WT 15000 by Ruff and Walker. In particular, the Gona pelvis might further complicate the ecogeographic story for thorax morphology.

RESPONSE 1: We agree with your valuable observation. Consequently, we have expanded the third paragraph of the Discussion to address your suggestion.

COMMENT 2: The method for estimating the position, size, and orientation of missing elements described on lines 285-297 raised some worries in my mind about how much error the procedure would produce in the resulting estimates. Can the authors provide a few more statistical details to clarify the magnitude of the uncertainty? I worry that the (very lovely) graphics in Figure 1 look authoritative but may conceal more uncertainty than they imply.

RESPONSE 2: We really appreciate your kind words about Figure 1. To begin with, all the details about the estimation of missing vertebral data can be found in García-Martínez et al. (2018), where the reconstruction methodology is thoroughly described and validated. Regarding ribs, we used the landmarks of 59 complete modern human ribcages to perform a Partial Least Squares (PLS) analysis per fossil ribcage. These PLS analyses had two blocks: all landmarks vs preserved costal landmarks of the target, which were selected ad hoc on the control sample depending on each fossil. The workflow started performing a GPA in form space per block. Then we aligned the case—that is, the (semi)landmarks of the partial fossil ribcage—to the block where we selected those preserved landmarks in the control sample. Next, we performed a PLS between the two blocks and predicted on one block (full ribcage) the case aligned to the other (partial ribcage) using the first latent variable of the analysis. The R script used for this procedure, which is based on package ‘Morpho’ v. 2.10, (Schlager, 2017), is attached below. The full R script including all the statistical analysis is available upon request to the corresponding author.

```
gpa_full <- procSym(full_thorax_control, CSinit = FALSE, scale=FALSE)
```

```
gpa_preserved <- procSym(selected_thorax_control, CSinit = FALSE, scale=FALSE)
```

```
gpa_fossil <- align2procSym(gpa_preserved,fossil)
test <- pls2B(gpa_full$orpdata,gpa_preserved$orpdata,rounds=999)
predict_case_fossil <- predictPLSfromData(test,y=gpa_fossil,ncomp = 1)
```

This explanation has been incorporated into the fourth paragraph of the Materials and Methods section to clarify the estimation method for readers.

References:

- D. García-Martínez, A. Riesco, M. Bastir, Missing element estimation in sequential anatomical structures: The case of the human thoracic vertebrae and its potential application to the fossil record, in Geometric Morphometrics: Trends in Biology, Paleobiology and Archaeology, C. Rissech, L. Lloveras, J. Nadal, J. M. Fullola, Eds. (SERP, Seminari d'Estudis i Recerques Preshistoriques, Universitat de Barcelona, Societat Catalana d'Arqueologia, 2018), pp. 93-97.
- S. Schlager, Morpho and Rvcg – Shape Analysis in R, in Statistical Shape and Deformation Analysis, G. Zheng, S. Li, G. Szekely, Eds. (Academic Press, London, UK, 2017), pp. 217-256.

Minor comments

COMMENT 3: Line 82: Please note that the new age for Jebel Irhoud reported by Hublin et al. is 315 ± 34 ka.

RESPONSE 3: Thank you for this comment. Date corrected.

COMMENT 4: Lines 123-134: Perhaps specify that Ohalo 2 bridges the gap in thoracic fossils between Skull/Qazeh and later modern humans in the Levant; there are additional, earlier cranial fragments of modern humans (e.g., the calvaria from Manot Cave).

RESPONSE 4: Good suggestion, you are completely right. We have clarified it in the text.

COMMENT 5: Line 639: Please change (Irak) to (Iraq).

RESPONSE 5: Thank you for pointing out that detail. Change done.

REVIEWER 2

Major comments

COMMENT 1: The fossil reconstructions of Nazlet Khater 2, Ohalo II H2, and Ötzi are based on 10 recent male *H. sapiens* specimens. By using the spatial arrangement of recent *H. sapiens* ribcages for the reconstructions, these reconstructions will inevitably inherit some of the characteristics of the recent *H. sapiens* ribcages, such as its globular proportions. The shape plot underscores this issue, as the only ribcage that stands out from the modern *H. sapiens* means is Dolní Věstonice 13. Would using complete fossil ribcages as templates for reconstructing the three distorted fossil *H. sapiens* specimens result in different reconstructions?

RESPONSE 1: This is an interesting question. First, it is indeed inevitable that the reference sample has some influence on the target when performing missing data estimation. However, it is important to clarify that the 10 recent male *H. sapiens* were used exclusively to estimate missing parts or elements of the fossil thoracic spine by vectors of serial change. This means that the influence of this control sample on the final 3D ribcage reconstruction was limited. Once the vertebral missing data were estimated, the thoracic spine of the fossils was reconstructed and their costal remains placed where it corresponded. Then, we used the 59 ribcages from the modern human control sample to estimate rib length in the three distorted fossil specimens, via 2B-PLS (explained above). This approach allowed us to preserve as much of the original rib anatomy and positioning as possible, minimizing bias and ensuring that the reconstructions respected the preserved fossil morphology.

COMMENT 2: Dolní Věstonice 13 clearly stands out from the other three fossil *H. sapiens* in shape space, while Nazlet Khater 2, Ohalo II H2, and Ötzi cluster more centrally on the shape plot (perhaps due to the reconstruction method as noted above). Hence, grouping Dolní Věstonice 13 and Ötzi as exhibiting ‘colder’ morphologies while classifying Nazlet Khater 2 and Ohalo II H2 as ‘warmer’ morphologies is not supported by the shape analysis. More justification for this grouping is needed beyond the brief mention in the results section that the Ötzi reconstruction displays a stockier and slightly higher rib declination.

RESPONSE 2: We completely agree with your interpretation. We have removed this grouping stating that Ötzi has mixed ribcage features (large size and globular shape), which would have been beneficial for seasonal alpine transhumance. This has been explained in depth in the second paragraph of the Discussion.

COMMENT 3: An image of the (semi)landmarks used in the analyses would be beneficial, along with a brief description of the (semi)landmark placement. This is particularly relevant given the discussion on reducing full ribcage anatomy to landmark data.

RESPONSE 3: To address this, we have added a new figure to the Supplementary Online Material (**Fig. S1**) showing the anatomical location and spatial distribution of all landmarks and semilandmarks on the ribcage. Additionally, we provide a brief description of each point through a new supplementary table (**Table S1**).

COMMENT 4: The introduction states that the Skhul and Qafzeh ribcages are sufficiently preserved, yet they are not included in this study. What is the reason for their exclusion?

RESPONSE 4: It is correct that the exclusion of these specimens is not appropriately explained in the text. We have now added a few lines to clarify this point. They were not included in our research for several reasons. First, although we planned to reconstruct the ribcages of Skhul IV and Qafzeh 9, we did not know whether it would have been possible since their costovertebral material is fragmentary, potentially deformed and still embedded in sediment. This would have made it difficult to obtain proper 3D models of the skeletal remains. In addition, their phylogenetic position is still under debate. Finally, we were unable to scan these fossil materials due to logistical and geopolitical issues beyond our control.

References:

- J. J. Shea, The middle Paleolithic of the East Mediterranean Levant. *J. World Prehist.* 17, 313-394 (2003).
- B. Vandermeersch, O. Bar-Yosef, The Paleolithic burials at Qafzeh cave, Israel. *PALEO* 30, 256-275 (2019).
- I. Hershkovitz, H. May, R. Sarig, A. Pokhojaev, D. Grimaud-Hervé, E. Bruner, C. Fornai, R. Quam, J. L. Arsuaga, V. A. Krenn, M. Martín-Torres, J. M. Bermúdez de Castro, L. Martín-Francés, V. Slon, L. Albessard-Ball, A. Vialet, T. Schöler, G. Manzi, A. Profico, F. di Vincenzo, G. W. Weber & Y. Zaidner. A Middle Pleistocene *Homo* from Neshar Ramla, Israel. *Science*, 372, 1424-1428 (2021).
- H. May, R. Sarig, A. Pokhojaev, C. Fornai, M. Martín-Torres, J. M. Bermúdez de Castro, G. W. Weber, Y. Zaidner, & I. Hershkovitz. Response to Comment on “A Middle Pleistocene *Homo* from Neshar Ramla, Israel”. *Science*, 374, eab15789 (2021).

Minor comments

COMMENT 5: Ohalo II H2 is categorised within the ‘warmer climate’ sample in line 142, yet the background information on lines 122-123 states that “climatic conditions in the Levant were cooler and more humid than they are today”. Additionally, Table 1 lists the climate for this specimen as temperate. Please clarify.

RESPONSE 5: We apologize for the misunderstanding. Our intention was to group the fossils into two broad categories: those from colder and those from warmer climates. Consequently, although the climate in which Ohalo II H2 lived was temperate, it was still warmer compared to Dolní Věstonice 13. For this reason, Ohalo II H2 was grouped with Nazlet Khater 2 in both the hypotheses and discussion.

COMMENT 6: The term ‘extant’ is used in Figure 1, line 302, and line 307 in reference to fossil *H. sapiens* specimens, which creates confusion since ‘extant’ typically refers to living species. Using ‘original’ vs ‘reconstructed’ material may perhaps provide greater clarity.

RESPONSE 6: You are correct that using the word ‘extant’ may lead to confusion when referring to fossil specimens. To enhance clarity, we have replaced ‘extant’ with ‘original’ throughout the manuscript.

COMMENT 7: Regarding the Nariokotome Boy, was the juvenile reconstruction or the estimated adult ribcage from Bastir et al. (2020) used in the analyses?

RESPONSE 7: This is an excellent question. During the experimental design, we were aware that it would be necessary to estimate missing data for our target sample (fossil ribcages of *Homo sapiens*). Therefore, we decided that the subsequent comparative analyses should be performed using original fossil material so we could reduce the bias of our results. This is why we did not incorporate the estimated adult ribcage of KNM-WT 15000

COMMENT 8: The centroid size results are well-written and clear. However, the analysis requires some more explanation in the methods section.

RESPONSE 8: Thank you for your appreciation. We have expanded the fourth paragraph of the Methods section to provide a brief explanation of what centroid size and UPGMA clustering represent.

COMMENT 9: Lines 213-215: The wording here is awkward and unclear. Please revise.

RESPONSE 9: Rewording done.

RESPONSE LETTER

REVIEWER 2

COMMENT 1: Firstly, I appreciate the authors' response regarding the use of modern human samples in fossil ribcage reconstructions. The clarification that recent *H. sapiens* data were used specifically for estimating missing vertebral elements and rib lengths, rather than for global ribcage morphology, is helpful. However, I believe the inherent limitations of the approach should be addressed somewhere in the manuscript. Even when used in a localised way, methods such as vectors of change and PLS embed structural assumptions derived from the modern human sample. These assumptions could subtly influence aspects of the reconstructed morphology, such as rib orientation, curvature, or thoracic volume, especially in cases where preservation is very limited. It would be helpful to explicitly acknowledge this in the discussion or methods section, along with the authors' efforts to preserve as much of the original fossil morphology as possible. Additionally, a brief discussion on whether the use of complete fossil ribcages as reference templates (where available) might yield meaningfully different reconstructions would be appropriate.

RESPONSE 1: Thank you for this thoughtful suggestion. We have expanded the limitations paragraph at the Discussion section in order to meet your requirements. However, we have not incorporated your final point due to the extremely limited availability of complete ribcages in the fossil record. While their use as reference templates would be ideal, it is currently unfeasible not only because such specimens are extremely rare and often inaccessible, but also because they would not constitute a sufficiently robust sample for generating reliable reconstructions.

COMMENT 2: Secondly, the authors' clarification concerning the inclusion of Ohalo II H2 in the "warmer climate" group provides helpful context. However, I still find this classification potentially confusing, especially given that the manuscript refers to the climate of Ohalo II H2 as "cooler and more humid than today" and labels it as "temperate" in Table 1. I suggest that the manuscript be revised to either (1) clearly explain the rationale for grouping Ohalo II H2 with the "warmer climate" specimens, despite its temperate classification, or (2) adjust the grouping terminology to avoid potential misinterpretation.

RESPONSE 2: You are completely right that Ohalo II H2 cannot be included within the group of warmer climates. According to Robinson et al. (2006), the estimated conditions for the Soreq Cave (near Jerusalem) ~23,000 years ago included mean annual temperatures of approximately 8–12 °C, and annual precipitation between 250 and 400 mm. These values are consistent with a cold semi-arid steppe climate (BSk) in the Köppen-Geiger classification (Peel et al., 2007). Consequently, we labeled Ohalo II H2 as "temperate". However, we grouped Ohalo II H2 within the "warmer climates" category in comparison to colder climates such as those represented by Dolní Věstonice 13. We acknowledge that this might have been confusing, so we have revised the text accordingly. Now, when referring to "warmer climates," we use the phrase "warmer and more temperate

climates” to better reflect the range of environmental conditions faced by Nazlet Khater 2 and Ohalo II H2, and to avoid potential misunderstandings. In addition, we have corrected the phrase “cooler and more humid than today” to “cooler and more arid than today”, as our initial description was mistaken.

References:

S. A. Robinson, S. Black, B. W. Sellwood, P. J. Valdes, A review of palaeoclimates and palaeoenvironments in the Levant and Eastern Mediterranean from 25,000 to 5000 years BP: setting the environmental background for the evolution of human civilisation. *Quat. Sci. Rev.* 25, 1517-1541 (2006).

M. C. Peel, B. L. Finlayson, T. A. McMahon, Updated world map of the Köppen-Geiger climate classification. *HESS* 11, 1633-1644 (2007).

COMMENT 3: Thirdly, since the reconstruction of KNM-WT 15000 used in the analyses represents a juvenile individual, this should be acknowledged in the discussion, especially when highlighting the proximity of Dolní Věstonice 13 to this specimen. The ribcage morphology of a juvenile may not reflect that of an adult, which could influence interpretations of morphological similarity. Given its immature age, it is also not surprising that KNM-WT 15000 has the smallest ribcage in the sample, and this should be acknowledged in the manuscript.

RESPONSE 3: Although we mentioned that KNM-WT 15000 is a juvenile individual in the Materials and Methods section, we agree that explicitly acknowledging this in the Discussion will help to clarify the interpretation of our results, improving the overall clarity and quality of the manuscript. Consequently, we have expanded the first paragraph of the Discussion section to address this comment.

COMMENT 4: Lastly, the phrase on line 175-176: “Fig. 1 depicts the extant and fully reconstructed thorax of Nazlet Khater 2, Dolní Věstonice 13, Ohalo II H2, and Ötzi” should be revised to replace “extant” with “original” in line with the rest of the revised manuscript terminology.

RESPONSE 4: Thank you for your observation, we overlooked this in the previous revision. Change done.